# Extremely Low-Frequency Electromagnetic Field (ELF-EMF) Increases Mitochondrial Electron Transport Chain Activities and Ameliorates Depressive Behaviors in Mice

**DOI:** 10.3390/ijms252011315

**Published:** 2024-10-21

**Authors:** Masaki Teranishi, Mikako Ito, Zhizhou Huang, Yuki Nishiyama, Akio Masuda, Hiroyuki Mino, Masako Tachibana, Toshiya Inada, Kinji Ohno

**Affiliations:** 1Division of Neurogenetics, Center for Neurological Diseases and Cancer, Nagoya University Graduate School of Medicine, Nagoya 466-8550, Japan; teranishi.masaki.u5@s.mail.nagoya-u.ac.jp (M.T.); vurt-huang@med.nagoya-u.ac.jp (Z.H.); nishiyama.yuki.k5@s.mail.nagoya-u.ac.jp (Y.N.); amasuda@med.nagoya-u.ac.jp (A.M.); 2Division of Material Science (Physics), Graduate School of Science, Nagoya University, Nagoya 464-8602, Japan; mino@bio.phys.nagoya-u.ac.jp; 3Department of Psychiatry, Nagoya University Hospital, Nagoya 466-8560, Japan; mtachibana@med.nagoya-u.ac.jp; 4Department of Psychiatry, Nagoya University Graduate School of Medicine, Nagoya 466-8550, Japan; inada@med.nagoya-u.ac.jp; 5Graduate School of Nutritional Sciences, Nagoya University of Arts and Sciences, Nisshin 470-0196, Japan

**Keywords:** extremely low-frequency electromagnetic field (ELF-EMF), social defeat stress, major depressive disorder, mitochondrial electron transport chain, oxidative stress

## Abstract

Compromised mitochondrial electron transport chain (ETC) activities are associated with depression in humans and rodents. However, the effects of the enhancement of mitochondrial ETC activities on depression remain elusive. We recently reported that an extremely low-frequency electromagnetic field (ELF-EMF) of as low as 10 μT induced hormetic activation of mitochondrial ETC complexes in human/mouse cultured cells and mouse livers. Chronic social defeat stress (CSDS) for 10 consecutive days caused behavioral defects mimicking depression in mice, and using an ELF-EMF for two to six weeks ameliorated them. CSDS variably decreased the mitochondrial ETC proteins in the prefrontal cortex (PFC) in 10 days, which were increased by an ELF-EMF in six weeks. CSDS had no effect on the mitochondrial oxygen consumption rate in the PFC in 10 days, but using an ELF-EMF for six weeks enhanced it. CSDS inactivated SOD2 by enhancing its acetylation and increased lipid peroxidation in the PFC. In contrast, the ELF-EMF activated the Sirt3-FoxO3a-SOD2 pathway and suppressed lipid peroxidation. Furthermore, CSDS increased markers for mitophagy, which was suppressed by the ELF-EMF in six weeks. The ELF-EMF exerted beneficial hormetic effects on mitochondrial energy production, mitochondrial antioxidation, and mitochondrial dynamics in a mouse model of depression. We envisage that an ELF-EMF is a promising therapeutic option for depression.

## 1. Introduction

Defective mitochondrial energy production and increased oxidative stress have been reported and reviewed in major depressive disorder (MDD) [1,2,3,4,5,6,7,8]. In humans, patients with MDD show decreases in the subunits of mitochondrial electron transport chain (ETC) complex I in the prefrontal cortex (PFC) [9]. Similarly, the expression of mitochondrial genes is dysregulated in the PFC in patients with MDD [10]. In addition, the mitochondrial ETC is a major source of reactive oxygen species (ROS). The production of excessive ROS introduces mutations in mitochondrial DNA (mtDNA) and lipid peroxidation of mitochondrial and plasma membranes [11], which subsequently compromises mitochondrial ETC activities [12]. Mitochondria have a defense system against ROS. Superoxide dismutase 2 (SOD2), the expression of which is regulated by sirtuin 3 (Sirt3) [13], converts superoxide radicals into hydrogen peroxide and is a key enzyme in the mitochondrial defense system against ROS. In the peripheral blood of MDD patients, SOD2 is decreased [14]; the oxidant/antioxidant ratio is increased [15]; myeloperoxidase is increased [16]; hydrogen peroxide and malondialdehyde are increased [4]; and oxidative damage to mtDNA is increased [17]. These observations point to the notion that MDD patients are subject to oxidative stress due to defective mitochondrial ETC activities. Conversely, over 50% of patients with mitochondrial diseases, who have low mitochondrial ETC activities due to pathogenic variants in mitochondrial or nuclear DNA, exhibit MDD [18,19]. In rodent brains, non-stressed mice that show anxiety-like behaviors in the light/dark box test or the elevated plus maze test have high levels of ROS in the cerebral cortex, hippocampus, and cerebellum [20] and low antioxidant levels in the cingulate cortex [21]. In addition, a mouse model of chronic mild stress, simulating depression, showed decreased mitochondrial antioxidant enzymatic activity in the frontal cortex, striatum, and hippocampus [22]; suppressed glycolysis and mitochondrial tricarboxylic acid cycle (TCA cycle) in the ventral hippocampus [23]; decreased mitochondrial ETC activities in the PFC, cortex, and cerebellum [24,25]; and decreased mitochondrial membrane potential/oxygen consumption in the cortex, hippocampus, and hypothalamus [26]. Similarly, a mouse model of chronic social defeat stress (CSDS) [27] simulating depression showed decreased mitochondrial ETC activities in the PFC, amygdala, hippocampus, and hypothalamus [28,29]; decreased mitochondrial membrane potential in the PFC [30,31]; a decreased amount of ATP in the PFC [29,30,32]; increased mtDNA mutations in the amygdala [33]; and mitochondrial loss in the amygdala [33]. Furthermore, strain-specific mtDNA variations in mice [34] and partial deficiency of NDUFS4 in mitochondrial ETC complex I generated by the International Gene Trap Consortium [35] worsened their anxiety-like behaviors. Thus, both in humans and rodents, mitochondrial defects are causally associated with MDD, and vice versa.

The adult human brain consumes up to 20% of oxygen inhaled, though it constitutes only 2% of our total body weight [36]. Glucose metabolism fuels 95% or more of the ATP production in the brain [37,38]. The ATP required by the neurons is mostly generated by the TCA cycle and the oxidative phosphorylation (OXPHOS) of glucose and lactate [38,39] and is provided by anaerobic glycolysis in astrocytes and oligodendrocytes [38,40]. Mitochondrial morphology, regulated by mitochondrial fission–fusion dynamics and mitophagy, has an impact on neuronal development [41,42], differentiation [43], and function [44,45]. Mutations in PINK1, a mitophagy-related protein, cause hereditary early-onset Parkinson’s disease [46]. In the *Drosophila Pink1* knockout model, mitochondrial respiration driven by the ETC is significantly reduced with decreased enzymatic activity of mitochondrial ETC complexes I and IV, and the reduced activity of ETC complexes I and IV caused by *Pink1* knockout is ameliorated by inserting an additional copy of *Drp1* encoding dynamin-related protein 1 (DRP1) [47]. In addition, CSDS induces mitochondrial fission in the PFC [29] and mitophagy in the hippocampus and amygdala [33,48]. Furthermore, the enhancement of mitochondrial metabolism through knockdown of DRP1 or injection of the peroxisome proliferator-activated receptor δ (PPAR-δ) agonist GW0742 into the PFC rescue the depressive-like behaviors induced by CSDS [29,30]. Thus, mitochondrial OXPHOS and mitochondrial dynamics play essential roles in the physiology of the brain.

We found that 4 ms pulses at a 10 µT intensity of an electromagnetic field that were repeatedly modulated from 1 to 8 Hz for 8 s, referred to as an extremely low-frequency electromagnetic field (ELF-EMF) herein, minimized the thermal hysteresis of modified Ringer’s solution [49]. We also showed that the ELF-EMF reduced the mitochondrial mass to 67% in 3 h and the mitochondrial membrane potential to 81% in 6 h in AML12 mouse hepatic cells [50]. Analysis of the mitochondrial ETC complexes showed that the ELF-EMF exclusively reduced ETC complex II to 89% in 8 min in mouse liver homogenates, indicating that the direct target of the ELF-EMF was likely to be a subunit constituting ETC complex II, although the exact target remained to be elucidated. The suppression of ETC complex II triggered mitophagy and subsequently induced the hormetic activation of PGC-1α-mediated mitochondrial biogenesis. We also examined various ELF-EMF conditions and found that the initial protocol most effectively reduced the mitochondrial mass in the AML12 cells [50]. Here, we examined the effects of an ELF-EMF on behaviors, mitochondrial ETC proteins, and activities in the brain and the oxidative stress responses in the brain in a mouse model of CSDS. We also showed that the ELF-EMF exerted beneficial hormetic effects on mitochondrial energy production, mitochondrial antioxidation, and mitochondrial dynamics in a mouse model of CSDS.

## 2. Results

### 2.1. The ELF-EMF Ameliorated Behavioral Deficits in the CSDS Mouse Model

We examined the effect of an ELF-EMF on a CSDS mouse model simulating depression (Figure 1A) [27]. The application of CSDS for 10 days made C57BL/6N mice stay away from an aggressor ICR mouse, which was represented by a decreased social interaction (SI) ratio. C57BL/6N mice with an SI ratio ≥ 1 were defined as “resilient”, and those with an SI ratio < 1 were defined as “susceptible” (Appendix A). Only the “susceptible” mice were used for the following experiments [27]. The “susceptible” mice were divided into two groups so that the average SI ratios were similar between the two groups (Appendix A). One of the two groups was exposed to the ELF-EMF (CSDS + ELF-EMF mice) (Figure 1B), whereas the other group was placed in the same environment without the ELF-EMF (CSDS mice). Control mice without CSDS were also separated into two groups so that the SI ratios were similar between the two groups (Appendix A). One of the two control groups was exposed to the ELF-EMF (ELF-EMF mice), whereas the other group was placed in the same environment without the ELF-EMF (control mice).

Two weeks after CSDS for 10 days, the mice showed behavioral despair, represented by increased immobility times in the tail suspension test (Figure 1C) and the forced swim test (Figure 1D), as have been reported previously [51,52,53]. Exposure to the ELF-EMF for two weeks mitigated the immobility times in both behavioral tests in the CSDS + ELF-EMF mice (Figure 1C,D). Six weeks after CSDS, the CSDS mice still spent less time in the interaction zone when the ICR mouse was in the wire-mesh cage compared to the control and ELF-EMF mice (Figure 1F). The ELF-EMF increased the time spent in the interaction zone in the presence of the ICR mouse (for CSDS + ELF-EMF mice) (Figure 1F). The plasma corticosterone levels six weeks after CSDS were not significantly different between the control, ELF-EMF, CSDS, and CSDS + ELF-EMF mice (*p* = 0.324 by one-way ANOVA). Taken together, the ELF-EMF ameliorated behavioral deficits in the CSDS mouse model.

### 2.2. CSDS Variably Decreased Mitochondrial ETC Proteins in the Prefrontal Cortex (PFC)

On the next day after CSDS, we analyzed the levels of mitochondrial proteins in the PFC, which is one of the essential cerebral regions associated with depression [51,54,55,56,57,58,59,60]. CSDS did not change the levels of the mitochondrial outer membrane protein VDAC1 in the PFC (Figure 2A,B). In contrast, among the mitochondrial ETC proteins, CSDS reduced UQCRC2 (complex III), UQCRFS1 (complex III), and COX-I (complex IV) and tended to reduce ATP5A (complex V) in the PFC (Figure 2C,D) but had no effect on NDUFB8 (a subunit of ETC complex I), SDHA (complex II), or SDHB (complex II). In addition, CSDS did not change the mitochondrial oxygen consumption rate (OCR), an indicator of mitochondrial ETC activities [61], in the cerebral cortex (Figure 2E–G). Taken together, CSDS decreased the levels of the subunits of ETC complexes III and IV but not of complexes I, II, and V or the mitochondrial outer membrane protein and had no effect on the OCR.

### 2.3. CSDS Induced the Acetylation of Mitochondrial SOD2 and Increased Proteins with Lipid Peroxidation in the PFC

We examined oxidative stress in the PFC next. On the next day after CSDS, the SOD2 protein levels were not changed, while its acetylation at lysine 68 (Ace-SOD2), which inactivates SOD2, was upregulated in the PFC (Figure 3C,D). Sirtuin 3 (Sirt3) deacetylates both forkhead box O3 (FoxO3a) and SOD2, and deacetylated FoxO3a increases the protein levels of SOD2 [13,62,63,64,65]. We found that CSDS did not change the protein levels of Sirt3 or FoxO3a (Figure 3A,B). CSDS also upregulated 4-hydroxynonenal (4-HNE)-modified protein levels, which are an indicator of cellular ROS-induced lipid peroxidation [11], in the PFC (Figure 3E,F). Increases in dihydropyridine and MitoSOX, markers of mitochondrial ROS, were also previously reported in the PFC in CSDS mice [29]. Thus, CSDS had no effect on the levels of Sirt3, FoxO3a, or SOD2 but enhanced the acetylation of SOD2 and lipid peroxidation.

### 2.4. The ELF-EMF Increased Mitochondrial ETC Proteins in the PFC

Six weeks after CSDS, the ELF-EMF increased the levels of proteins of mitochondrial ETC complexes I (NDUFB8), II (SDHA and SDHB), III (UQCRC2 and UQCRFS1), and IV (COX-I) in the CSDS + ELF-EMF mice, although statistical significance was not observed for UQCRC2 (Figure 2J,K). Exposure to the ELF-EMF in the control mice (ELF-EMF mice) also tended to increase the levels of proteins of mitochondrial ETC complexes I (NDUFB8), II (SDHA and SDHB), and IV (COX-I) but not III (UQCRFS1 and UQCRC2) (Figure 2J,K). In contrast, the ELF-EMF had no effect on the protein levels of mitochondrial ETC complex V (ATP5A) (Figure 2J,K) or mitochondrial outer membrane proteins (VDAC1) (Figure 2H,I) in either the ELF-EMF or CSDS + ELF-EMF mice.

### 2.5. The ELF-EMF Increased the Mitochondrial OCR in the Cerebral Cortex

We previously showed that continuous ELF-EMF exposure for four weeks in wild-type mice increased the mitochondrial OCR in the liver [50]. Six weeks after CSDS, the ELF-EMF increased ADP-induced respiration (Phase II) and the spare respiratory capacity (Phase IV–Phase I) in the OCR assay in the cerebral cortex in the CSDS + ELF-EMF mice (Figure 2L–N). These results were consistent with the increased protein levels of mitochondrial ETC complexes I to IV in the PFC.

### 2.6. The ELF-EMF Activated the Sirt3-FoxO3a-SOD2 Pathway and Reduced CSDS-Induced Lipid Peroxidation

We next examined whether applying the ELF-EMF for six weeks activated the Sirt3-FoxO3a-SOD2 pathway due to an enhanced OCR and a subsequent increase in ROS production. In both the ELF-EMF and CSDS + ELF-EMF mice, the ELF-EMF increased the protein levels of Sirt3 and FoxO3a (Figure 3G,H). In the CSDS + ELF-EMF mice, the ELF-EMF also increased SOD2, while the acetylation of SOD2 remained unchanged (Figure 3I,J). In contrast, in the ELF-EMF mice, SOD2 was marginally increased (Figure 3I,J), but the reason for the lack of the effect on SOD2 acetylation remains unknown. As stated above, CSDS increased the 4-HNE-modified protein levels one day after CSDS (Figure 3E,F), which disappeared 6 weeks after CSDS (Figure 3K,L). We observed that the ELF-EMF decreased the 4-HNE-modified protein levels 6 weeks after CSDS (Figure 3K,L). Thus, ELF-EMF-mediated increased mitochondrial ETC activities were likely to have activated the Sirt3-FoxO3a-SOD2 pathway, and lipid peroxidation was suppressed.

### 2.7. CSDS Upregulated Fission-Promoting Phosphorylation of DRP1, Which Was Not Changed by ELF-EMF Exposure

We next analyzed markers for the mitochondrial fission–fusion dynamics [66,67]. On the next day after CSDS, phosphorylation at serine 616 of DRP1, a marker for mitochondrial fission, as well as the protein levels of mitochondrial dynamin-like GTPase (OPA1) and mitofusin 1 (MFN1), which are markers for mitochondrial fusion, remained unchanged in the PFC (Appendix A). In contrast, six weeks after CSDS, phosphorylation of DRP1 was increased in the PFC in the CSDS mice, indicating the activation of mitochondrial fission (Figure 4A,D–F). In contrast, CSDS did not change the protein levels of OPA1 and MFN1, which indicated that CSDS had no effect on mitochondrial fusion (Figure 4A–C). The ELF-EMF had no effect on the enhanced phosphorylation of DRP1 by CSDS or on the protein levels of OPA1 and MFN1 (Figure 4A–F).

### 2.8. CSDS Induced Mitophagy, Which Was Suppressed by the ELF-EMF

We next examined markers for mitophagy in the PFC. Six weeks after CSDS, CSDS had increased the protein levels of PINK1, a mitophagy-related protein, and the LC3-II/LC3-I ratio, a marker for mitophagy activation, in the PFC (CSDS mice), while the ELF-EMF suppressed these increases (CSDS + ELF-EMF mice) (Figure 4G,H). Thus, the ELF-EMF mitigated CSDS-induced enhanced mitophagy in the PFC.

## 3. Discussion

We recently reported that an ELF-EMF specifically suppresses mitochondrial ETC complex II activity in mouse liver homogenates and induces mitophagy in cultured cells, which subsequently triggers hormetic mitochondrial biogenesis and increases mitochondrial energy production [50]. As stated in the introduction, decreased mitochondrial ETC proteins and decreased mitochondrial ETC activities have been reported in depression in rodents [24,25,28,30,68] and humans [9,69]. We thus examined the effects of an ELF-EMF on a CSDS mouse model recapitulating depression. Rodent models for depression include chronic unpredictable mild stress, physical pain, learned helplessness, chronic restraint stress, lipopolysaccharide-induced stress, and CSDS [70,71]. Among them, CSDS etiologically mimics the pathophysiology of depression in humans [27,72,73] because exposure to chronic stress is a significant risk factor for the development of depression [74,75].

We first showed that CSDS variably decreased the protein levels of mitochondrial ETC proteins in the PFC (Figure 2C,D) but had no effect on the mitochondrial OCR in the cerebral cortex (Figure 2E–G). The difference in the brain samples (the PFC or the cerebral cortex) was likely to account for the differential mitochondrial features. Decreased gene expression related to mitochondrial functions in the medial PFC is shared between human MDD patients and CSDS-susceptible mice [76]. In contrast to this report and our current study, a mouse model of depression due to chronic restraint stress showed increased expression of mitochondrial ETC genes in the PFC [77]. This discrepancy may be accounted for by the differences in the durations and the stresses applied. The Sirt3-FoxO3a-SOD2 pathway is an essential mitochondrial defense mechanism against ROS [13,62], which contributes to a decrease in lipid peroxidation [78]. We showed that CSDS did not change the protein levels of Sirt3, FoxO3a, or SOD2 but increased the acetylation of SOD2 in the PFC (Figure 3A–D), which reduced the enzymatic activity of SOD2 [79]. CSDS also increased protein modification by 4-HNE (Figure 3E,F), a marker for lipid peroxidation [78]. In patients with MDD, SOD2 was decreased in their peripheral blood [14], and the 4-HNE levels were increased in their plasma [80]. Similarly, in patients with coronary artery disease with or without depression, serum 4-HNE levels were increased in patients with depression compared to those without depression [81]. Thus, reduced SOD2 activity, which makes the neuronal and glial cells vulnerable to oxidative stress, and increased 4-HNE levels are likely to be shared features in CSDS mice and MDD patients.

We next showed that exposure to the ELF-EMF for six weeks ameliorated depressive behaviors in the CSDS mouse model (Figure 1F). The ELF-EMF increased mitochondrial ETC complexes I to IV but not complex V in the PFC (Figure 2J,K) and increased the mitochondrial OCR (Figure 2L–N) in the CSDS + ELF-EMF mice. We also showed that the ELF-EMF increased the protein levels of Sirt3 and FoxO3a in the CSDS + ELF-EMF mice (Figure 3G,H). Furthermore, the ELF-EMF increased the protein levels of SOD2, while the acetylation of SOD2 remained unchanged, in the CSDS + ELF-EMF mice (Figure 3I,J). An increased mitochondrial OCR is expected to increase the production of ROS, which subsequently activate the Sirt3-FoxO3a-SOD2 pathway [82,83]. SDHA in mitochondrial ETC complex II is the deacetylation target of Sirt3 in mitochondrial ETC complexes [84,85]. Indeed, the overexpression of Sirt3 enhances mitochondrial ETC complex II activity [85]. Thus, the ELF-EMF facilitated a positive feedback loop between the enhanced mitochondrial OCR and the activated Sirt3-FoxO3a-SOD2 pathway in the CSDS + ELF-EMF mice. Since Sirt3 and the subsequent induction of SOD2 have antioxidant effects [78], the reduction in proteins modified by 4-HNE in the CSDS + ELF-EMF mice might have been due to the upregulation of the Sirt3-FoxO3a-SOD2 pathway. Similar to the findings of our studies, Sirt3 has been shown to play a critical role in the antidepressant and anxiolytic-like effects of kaempferol [86] and nicotinamide mononucleotide [87]. In addition, mitochondrial activation by Sirt3 was neuroprotective in a cell model of Parkinson’s disease [88,89]. The amelioration of behavioral deficits in the CSDS mouse model is thus likely to have been at least partly accounted for by increased Sirt3.

We recently showed that the ELF-EMF increased HSP70 in AML12 and HEK293 cells [90]. As HSP70 induces Sirt3 [91], Sirt3 might have been induced via the activation of HSP70. Alternatively, as Sirt3 directly deacetylates SDHA and enhances its activity [84], the inhibition of succinate dehydrogenase by the ELF-EMF might have induced the expression of Sirt3 in compensation and subsequently deacetylated and activated SDHA.

We showed that CSDS increased a marker for mitochondrial fission (the phosphorylation of DRP1) but not for fusion (protein levels of MFN1 or OPA1) in the PFC (Figure 4A–F). We also showed that CSDS increased markers for mitophagy (protein levels of PINK1 and the LC3-II/LC3-I ratio), which selectively degrades defective mitochondria with a low mitochondrial membrane potential [92,93] (Figure 4G,H). When axons are injured, DRP1-dependent mitochondrial fission is required to prevent neuronal cell death and axonal degeneration [94]. Although there was no evidence of axonal injury in the CSDS mouse model, CSDS decreases dendritic spines in the PFC [95], which may require the activation of mitochondrial fission by DRP1. In ischemic–hypoxic stress in rat brains, DRP1 triggers mitophagy [92,93]. Similarly, in African green monkey kidney fibroblast-like COS7 cells, DRP1-mediated mitochondrial fission stimulates both mitophagy and mitochondrial biogenesis [96]. We showed that the ELF-EMF suppressed CSDS-induced mitophagy (Figure 4G,H) but had no effects on the enhanced mitochondrial fission (Figure 4A,D,E,F). Thus, the enhanced mitochondrial energy production by the ELF-EMF in the CSDS mouse model was likely to have obviated the activation of mitophagy, although the ELF-EMF failed to suppress the activated mitochondrial fission.

Therapeutic effects of electromagnetic fields have been reported in mouse models of Alzheimer’s disease [97,98,99,100] and bone fragility [101], as well as in a rat model of neuropathic pain [102], to name a few. Among these, the enhancement of mitochondrial function by an electromagnetic field was reported in a mouse model of Alzheimer’s disease [97]. However, the effects of electromagnetic fields are not solely limited to enhanced mitochondrial activity. The reported targets of electromagnetic fields include calcium homeostasis, cryptochrome-based radical pairs, ROS levels, cell proliferation, and others [103]. We have also reported that an ELF-EMF activates heat shock proteins by enhancing their acetylation [90]. Therefore, enhanced mitochondrial activity is unlikely to have been the sole mechanism of the effects of the ELF-EMF in our CSDS mouse model.

Repetitive transcranial magnetic stimulation (rTMS), which generates electromagnetic pulses of more than 1 T, is one of the approved therapeutic modalities for patients with depression [104]. rTMS decreases oxygenized hemoglobin in the right dorsolateral prefrontal cortex in healthy subjects [105]. As activated mitochondria consume more oxygen, the therapeutic mechanisms of rTMS may partly be accounted for by mitochondrial activation, as we observed in using the ELF-EMF. The electromagnetic intensity of 10 µT that we used in the current study was 62.5 times less than the exposure threshold for the general public according to the guidelines by the International Commission on Non-Ionizing Radiation Protection (ICNIRP) [106]. We hope that using an ELF-EMF becomes an alternative therapeutic option for depression, especially for the ~30% of patients who do not respond to antidepressants [107]. As the enhancement of mitochondrial ETC activity is expected to ameliorate Alzheimer’s disease [108] and Parkinson’s disease [109], ELF-EMFs may also exert therapeutic effects on these neurodegenerative diseases.

## 4. Materials and Methods

### 4.1. Mouse Studies

All procedures were approved by the Animal Care and Use Committee of Nagoya University and were carried out in accordance with the relevant guidelines. Wild-type C57BL/6N male mice (6–7 weeks of age) and wild-type ICR male mice (10 weeks of age) were purchased from CLEA Japan and Japan SLC, respectively. The mice were housed in a controlled room (temperature, 20 to 25 °C; humidity, 40 to 70%; light/dark cycle, 12 h/12 h, 8 a.m./8 p.m.) with access to food and water ad libitum.

### 4.2. Protocols of Chronic Social Defeat Stress (CSDS)

CSDS as previously described [27] was employed with a minor modification. As previously reported [110], a C57BL/6N male mouse was placed in a cage in which an ICR male mouse, 10 weeks old or older, was housed. When the ICR mouse started to attack the C57BL/6N mouse in one minute on at least two out of three consecutive days, the ICR mouse was defined to be aggressive and was used in the following study. A C57BL/6N mouse was exposed to an aggressive ICR mouse for 5–10 min [110]. During the exposure, the attacked C57BL/6N mouse showed avoidance and fear behaviors. After the confrontation, they were separated by a perforated Plexiglas divider for the remainder of the 24 h period. During the separation, the C57BL/6N mouse smelled and saw the aggressor ICR mouse through the divider. The pair of the C57BL/6N mouse and the aggressor ICR mouse was randomly changed every day. To establish control C57BL/6N mice without CSDS, a C57BL/6N mouse was substituted for the aggressor ICR mouse. The pair of C57BL/6N mice was randomly changed every day. The social defeat stress was applied for 10 days, and the stressed and control mice were housed individually after these 10 days (Figure 1A).

### 4.3. Behavioral Tests

All behavioral tests were performed from 12 pm to 6 pm. The mice were habituated to the room for the behavioral tests 1 h before testing. For the purpose of screening, we performed the social interaction test, the tail suspension test, and the forced swim test in some mice 2 and 6 weeks after CSDS. In the social interaction test, the effect of the ELF-EMF was prominent 6 weeks after CSDS. In the tail suspension test and the forced swim test, the effects of CSDS disappeared 6 weeks after CSDS. We thus performed the tail suspension test and the forced swim test 2 weeks after CSDS and the social interaction test 6 weeks after CSDS (Figure 1A).

### 4.4. The Social Interaction Test

The social interaction test comprised two sessions [27]. Briefly, a C57BL/6N mouse was placed in a box (40 cm width × 50 cm depth × 32 cm height) containing a removable empty wire-mesh cage (6 cm width × 14 cm depth × 40 cm height) positioned against the wall and allowed to explore freely. The mouse was video-recorded under unmanned conditions. In the first session, the experimental mouse was placed in the box without an ICR male mouse in the wire-mesh cage for 15 min, and the time it spent in the interaction zone (defined as the 8 cm wide area surrounding the wire-mesh cage) in the last 150 s was recorded. In the second session, an ICR male mouse that had not been used to apply CSDS was placed in the wire-mesh cage, and the time spent in the interaction zone by the C57BL/6N mouse in the 150 s was recorded (Figure 1E). The social interaction ratio (SI ratio) was calculated by dividing the time it spent in the interaction zone with the ICR mouse by that without the ICR mouse. The social interaction test was performed the day after the 10-day stress period and 6 weeks after the stress period. In the first social interaction test, mice with an SI ratio < 1 were defined as “susceptible” to CSDS, whereas those with an SI ratio ≥ 1 were defined as “resilient” and were excluded from the subsequent analysis.

### 4.5. The Tail Suspension Test

Each experimental mouse was suspended 40 cm above the floor by fixing its tail using adhesive tape 1 cm from the tail tip for 10 min. The mouse’s movements were video-recorded under unmanned conditions. The immobility time of each mouse in the 10 min was obtained.

### 4.6. The Forced Swim Test

A plastic cylinder (16 cm diameter × 20 cm height) was filled with water up to 12 cm above the bottom. The water was left at room temperature for more than 2 h to set the water temperature to 25 °C before the experiment. Each mouse was placed in the water for 6 min. The mouse’s movements were video-recorded under unmanned conditions. The immobility time of each mouse in the last 4 min was obtained. The forced swim test was performed 2 h after the tail suspension test. Although conducting forced swim and tail suspension tests serially in a day has previously been reported [111,112,113,114], the serial tests might have caused additional stress.

### 4.7. Preparation of Brain Samples for Western Blotting

Each mouse was sacrificed by decapitation the day after the second social interaction test. Its prefrontal cortex (PFC) was immediately dissected out of the brain on an ice-cold plate. Each PFC sample was quickly weighed, frozen by dry ice in a 1.5 mL micro tube, and kept in a deep freezer at −80 °C. Frozen PFC samples were defrosted in RIPA buffer (cat#89901, Thermo Fisher Scientific, Waltham, MA, USA) containing 1 μg/µL aprotinin, 1 μg/µL leupeptin, 1 μg/µL pepstatin A, 1 mM PMSF, and a Phosphatase Inhibitor Cocktail (PhosSTOP, Roche, Basel, Switzerland) and then homogenized using sonication with a Sonifier SFX550 and a Sonifier Cell Disruptor 1″ Diameter High-Intensity Cup Horn at 60% pulse strength for 2 min (cat#101-147-046, Emerson, St. Louis, MO, USA). After homogenization, the lysates were incubated at 4 °C for 20 min and centrifuged at 12,600× *g* at 4 °C for 20 min. The supernatant was incubated at 95 °C for 5 min in sample buffer (62.5 mM Tris-HCl, pH 6.8; 2% SDS; 10% glycerol; 0.005% bromophenol blue; and 2% 2-mercaptoethanol).

### 4.8. Western Blotting

The supernatant in the sample buffer was separated by electrophoresis on 10%, 12%, 13%, 14%, or 16% SDS–polyacrylamide gel and blotted onto a polyvinylidene fluoride membrane (Immobilon-P membrane, IPVH00010, Millipore, Burlington, MA, USA). The membranes were washed in Tris-buffered saline (pH 7.6) containing 0.05% Tween 20 (TBS-T) and blocked for 30 min at room temperature in TBS-T with 5% milk (sc-2325, Santa Cruz Biotechnology, Dallas, TX, USA). The membranes were reacted overnight at 4 °C with the primary antibodies shown in Appendix A. The membranes were washed with TBS-T and incubated with a secondary horse anti-mouse IgG (1:2000, cat#7076, Cell Signaling Technology, Danvers, MA, USA) or goat anti-rabbit IgG (1:2000, cat#7074, Cell Signaling Technology) antibody conjugated to horseradish peroxidase (HRP) for 1 h at room temperature. The bound antibodies were detected with Amersham ECL Western blotting detection reagents (GE Healthcare, Chicago, IL, USA) and an ImageQuant LAS 4000 mini (GE Healthcare). The signal intensity was quantified using ImageQuant TL software 8.2 (GE Healthcare).

### 4.9. Enzyme-Linked Immunosorbent Assay (ELISA)

Blood was collected from the decapitated mouse bodies. Heparin was quickly added to each blood sample, and this was then centrifuged at 1500× *g* for 10 min at 4 °C. Plasma corticosterone levels were determined using the Corticosterone ELISA kit (ADI-900-097, Enzo Life Sciences, Farmingdale, NY, USA) according to the manufacturer’s instructions.

### 4.10. Mitochondrial Isolation for OCR Measurements

As the measurement of all mitochondrial ETC activities requires a large number of brain samples, we measured mitochondrial ETC activities according to the OCR. Brain mitochondria were isolated by differential centrifugation, as described previously [115,116]. Briefly, following decapitation, the cerebral cortex was dissected out and homogenized using a glass Dounce homogenizer in MAS buffer (220 mM mannitol, 10 mM KH_2_PO_4_, 5 mM MgCl_2_, 1 mM EGTA, 70 mM sucrose, 2 mM HEPES, and 0.2% (*w*/*v*) BSA, pH 7.3). The homogenates were centrifuged at 800× *g* for 8 min at 4 °C. The supernatant was centrifuged again at 800× *g* for 4 min at 4 °C. Then, the supernatant was further centrifuged at 8000× *g* for 10 min at 4 °C. The pellets containing the mitochondria were resuspended in BSA-free MAS buffer (220 mM mannitol, 10 mM KH_2_PO_4_, 5 mM MgCl_2_, 1 mM EGTA, 70 mM sucrose, and 2 mM HEPES, pH7.3). The protein concentration was determined using Pierce 660 nm protein assay reagent (cat#22660, Thermo Fisher Scientific) according to the manufacturer’s instructions.

### 4.11. Measurement of the Oxygen Consumption Rate (OCR) of Isolated Mitochondria

The OCR was measured in MAS buffer using a Seahorse XFe24 Analyzer (Agilent Technologies, Santa Clara, CA, USA). The freshly isolated mitochondria (15 µg of protein) described above were plated into each well of an XF24 V7 PS plate (cat#100777-004, Agilent Technologies) in 50 µL of MAS buffer. In order to attach the mitochondria to the plate, the plate was centrifuged at 1500× *g* for 5 min at room temperature. Then, 450 µL of MAS buffer containing 10 mM succinic acid, 10 mM malic acid, 10 mM glutamic acid, and 20 mM sodium pyruvate was added to each well. First, the basal OCR was measured (Phase I). Next, ADP (5 mM) was injected (Phase II), followed by sequential injections of oligomycin (10 µM) (Phase III), FCCP (5 µM) (Phase IV), and antimycin A/rotenone (both 5 µM) (Phase V). Serial injections enabled the determination of the ADP-induced respiration (Phase II), proton leak (Phase III), maximal respiration (Phase IV), and non-mitochondrial oxygen consumption (Phase V), respectively.

### 4.12. The ELF-EMF Apparatus

The ELF-EMF was applied to the mice using a rectangular coil (18 cm width × 28 cm depth, 60 turns of copper wire of a 0.32 mm diameter) around a mouse’s cage. The current controller generated a ~10 µT electromagnetic field of a 4 ms pulse width with increasing frequencies of 1, 2, 3, 4, 5, 6, 7, and 8 Hz for 8 s (ELF-EMF) (Figure 1B). The ~10 µT magnetic field was generated at the level of the mouse’s head, which was 2 cm above the floor (Figure 1B). Before and after each experiment, we confirmed the intensity of the magnetic flux using an EMF tester (cat#RT-100, Erickhill).

### 4.13. Statistical Analysis

The normality of the data distribution was first examined using the Shapiro–Wilk test, followed by Bonferroni correction, and all the datasets followed a normal distribution with adjusted *p*-values > 0.10. Statistical significance was calculated using Student’s *t*-test, one-way ANOVA, or two-way ANOVA, followed by Šídák’s, Tukey’s, or Dunnett’s post hoc test, using GraphPad Prism 9.4.1. The choice of the post hoc test was in accordance with the recommendation by GraphPad Prism. Statistical significance was set to *p* < 0.05.

## 5. Conclusions

We showed that the ELF-EMF ameliorated CSDS-induced behavioral changes. The ELF-EMF enhanced the mitochondrial ETC activity and the Sirt3-FoxO3a-SOD2 pathway and suppressed lipid peroxidation and mitophagy in the PFC in the CSDS mice. Our findings underscore previously reported observations that mitochondrial ETC activities are decreased in depression and also imply that their enhancement is likely to ameliorate the biochemical and behavioral defects in depression.

## Figures and Tables

**Figure 1 ijms-25-11315-f001:**
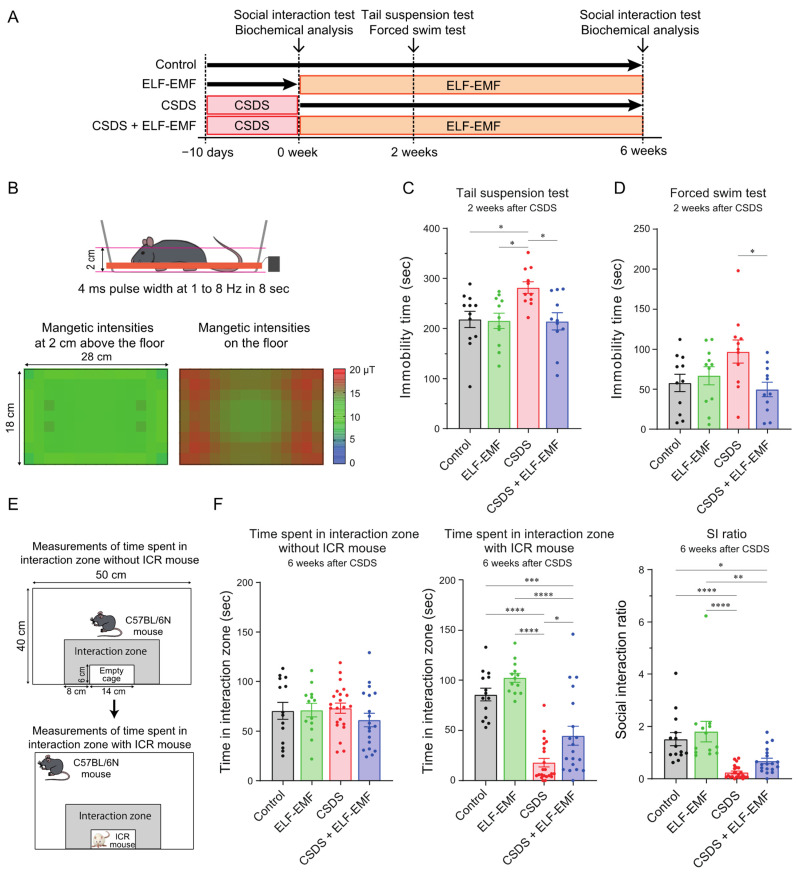
The ELF-EMF exerted antidepressant-like behavioral effects in the CSDS mouse model of depression. (**A**) Experimental protocols: C57BL/6N mice were divided into four groups. CSDS was applied for 10 continuous days. CSDS-resilient mice were excluded from the following studies. CSDS-susceptible mice were individually housed in a cage with or without the ELF-EMF for six weeks. (**B**) Electromagnetic intensities were generated by a coil around the mouse cage. The mouse’s head was about 2 cm above the floor. (**C**) Immobility times in the tail suspension test two weeks after CSDS (*n* = 11 to 12 mice each). (**D**) Immobility times in the forced swim test two weeks after CSDS (*n* = 11 mice each). (**E**) Schematic of the social interaction test. (**F**) Time spent in the interaction zone with or without an ICR mouse and the SI ratio, which is calculated by dividing the time spent with the ICR mouse by that without the ICR mouse six weeks after CSDS (*n* = 14, 13, 22, and 19 mice, respectively). (**C**,**D**,**F**) Mean and SEM are indicated. * *p* < 0.05, ** *p* < 0.01, *** *p* < 0.001, and **** *p* < 0.0001 according to one-way ANOVA, followed by Tukey’s multiple comparisons test.

**Figure 2 ijms-25-11315-f002:**
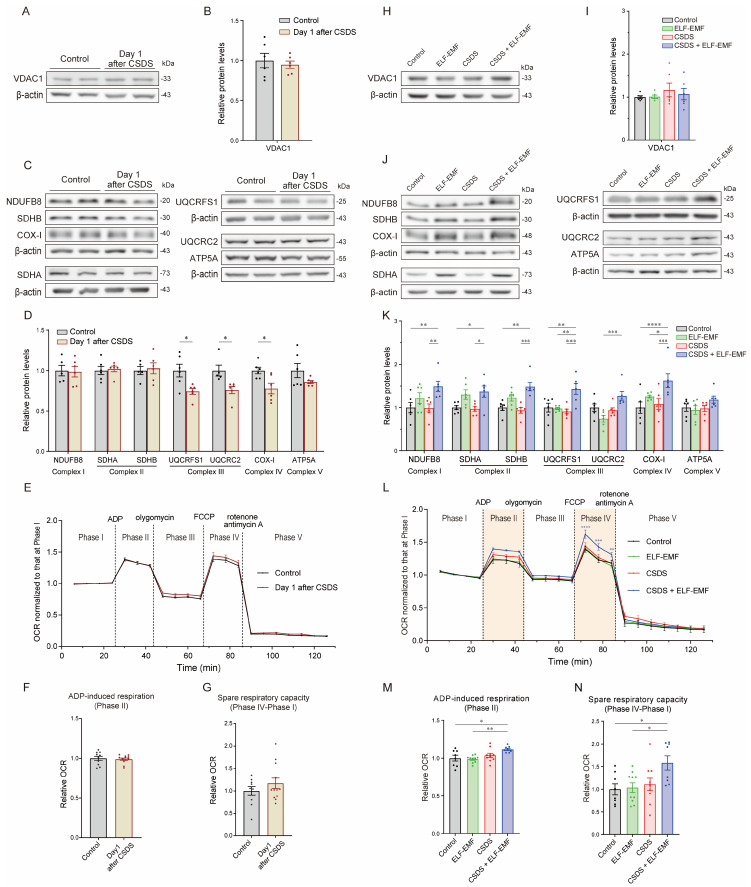
CSDS for 10 consecutive days variably downregulated the mitochondrial electron transport chain (ETC) complex proteins in the prefrontal cortex (PFC), which were increased by the ELF-EMF for 6 weeks. Spare respiratory capacity according to the OCR remained unchanged by CSDS but was increased by using the ELF-EMF for 6 weeks. (**A**,**C**) Representative duplicated immunoblots of the PFC on the next day after CSDS. (**B**,**D**) Densitometric analysis of immunoblots. Mean and SEM are indicated (*n* = 5 to 6 mice each). (**B**) No statistical difference according to the unpaired *t*-test. (**D**) * *p* < 0.05 according to two-way ANOVA, followed by Šídák’s multiple comparisons test. (**E**) The OCR of mitochondria isolated from the cerebral cortex on the next day after CSDS. The OCR was normalized to that at Phase I. Mean and SEM are indicated (*n* = 10 and 12 brain hemispheres, respectively). No statistical difference by two-way repeated-measures ANOVA. (**F**,**G**) ADP-induced respiration (Phase II) (**F**) and spare respiratory capacity (Phase IV–Phase I) (**G**) of mitochondria isolated from the cerebral cortex on the next day after CSDS are normalized to the mean of the control. Mean and SEM are indicated (*n* = 10 and 12 brain hemispheres, respectively). No statistical difference according to the unpaired *t*-test. (**H**,**J**) Representative immunoblots of the PFC six weeks after CSDS. (**I**,**K**) Densitometric analysis of immunoblots. Mean and SEM are indicated (*n* = 6 mice each). (**I**) No statistical difference according to one-way ANOVA. (**K**) * *p* < 0.05, ** *p* < 0.01, *** *p* < 0.001, and **** *p* < 0.0001 according to two-way ANOVA, followed by Tukey’s multiple comparisons test. (**L**) The OCR of mitochondria isolated from the cerebral cortex. The OCR was normalized to that at Phase I. Mean and SEM are indicated (*n* = 8 to 10 brain hemispheres each). Statistical significance was observed in CSDS + ELF-EMF compared to CSDS according to two-way repeated-measures ANOVA, followed by Dunnett’s multiple comparisons test (** *p* < 0.005, *** *p* < 0.0005, and **** *p* < 0.0001). (**M**,**N**) ADP-induced respiration (Phase II) (**M**) and spare respiratory capacity (Phase IV–Phase I) (**N**) of mitochondria isolated from the cerebral cortex are normalized to the mean of the control. Mean and SEM are indicated (*n* = 8 to 10 brain hemispheres each). * *p* < 0.05 and ** *p* < 0.01 according to one-way ANOVA, followed by Tukey’s multiple comparisons test.

**Figure 3 ijms-25-11315-f003:**
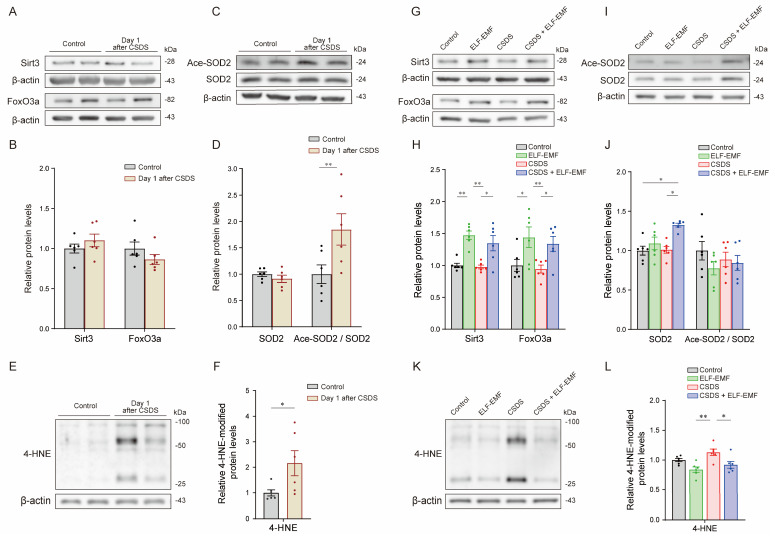
CSDS for 10 consecutive days increased the acetylation at K68 of SOD2 and the levels of 4-HNE-modified proteins in the prefrontal cortex (PFC), which were mostly mitigated by ELF-EMF exposure for 6 weeks. (**A**,**C**,**E**) Representative duplicated immunoblots of the PFC on the next day after CSDS. (**B**,**D**) ** *p* < 0.01 according to two-way ANOVA, followed by Šídák’s multiple comparisons test. (**F**) * *p* < 0.05 according to the unpaired *t*-test. (**G**,**I**,**K**) Representative immunoblots of the PFC six weeks after CSDS. (**H**,**J**,**L**) Densitometric analysis of immunoblots. Mean and SEM are indicated (*n* = 6 mice each). (**H**,**J**) * *p* < 0.05 and ** *p* < 0.01 according to two-way ANOVA, followed by Tukey’s multiple comparisons test. (**L**) * *p* < 0.05 and ** *p* < 0.01 according to one-way ANOVA, followed by Tukey’s multiple comparisons test.

**Figure 4 ijms-25-11315-f004:**
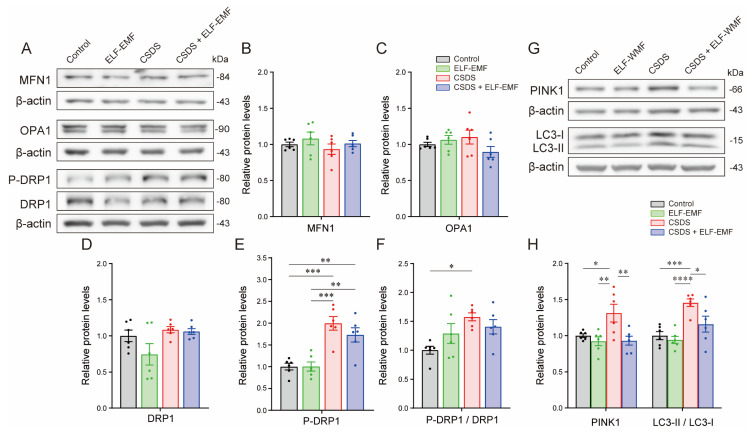
CSDS-mediated enhanced phosphorylation of DRP1 in the PFC was not changed by the ELF-EMF, whereas CSDS-mediated upregulation of mitophagy in the PFC was mitigated by the ELF-EMF six weeks after CSDS. (**A**,**G**) Representative immunoblots six weeks after CSDS. (**B**–**F**,**H**) Densitometric analysis of immunoblots. Mean and SEM are indicated (*n* = 6 mice each). (**B**–**F**) * *p* < 0.05, ** *p* < 0.01, and *** *p* < 0.001 according to one-way ANOVA, followed by Tukey’s multiple comparisons test. (**H**) * *p* < 0.05, ** *p* < 0.01, *** *p* < 0.001, and **** *p* < 0.0001 according to two-way ANOVA, followed by Tukey’s multiple comparisons test.

## Data Availability

Original immunoblot images are available in Appendix A. No OMICS data were generated in the current study.

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
