# Peer review of "Extremely Low-Frequency Electromagnetic Field (ELF-EMF) Increases Mitochondrial Electron Transport Chain Activities and Ameliorates Depressive Behaviors in Mice"

_ijms, 2024, doi:10.3390/ijms252011315_

Round 1

Reviewer 1 Report

Comments and Suggestions for Authors

In this manuscript the authors investigated “the effects of 90 ELF-EMF on behaviors, mitochondrial ETC proteins and activities in the brain, and oxidative stress responses in the brain in a mouse model of CSDS”. Their results show a beneficial effect on a number of mitochondrial functions.

According to the methods, the aggressor mouse was 10 weeks and older compared to 6-7 weeks of experimental mice. My question is whether the aggressor mouse is of the same strain or not? If not, why was a different mouse strain chosen, as this alone is already a significant stress factor for the mice?

Line 102: Is here meant, that both groups of mice have a similar SI ratio? If yes, please change “became” to “were”.  Please consider the same in line 106.

Figure 1A shows which behavioral and biochemical experiment was done at a certain time point. Why where not all experiments done for all time points? It would be interesting to see the changes of behavior over time.

Figure 1C and D: this is an amazing result. I would have expected that already stressed and depressed mice want to get away from a stressful environment as quickly and as soon as possible. However, it appears that the mice gave up and adapted to a stressful situation more easily.

Please add in Figure 1, which results is from 1 day, 2 weeks or 6 weeks after CSDS.

Figure 2 and 4: please combine the results of these figures. They show the same experiments for 2 different time points. Besides the expression levels of the ETC proteins, it would be interesting to see their enzymatic activity. Expression may not change, but modification by acetylation, as the authors show in figure 4C, may change the activity.

Line 202: typo “sated” instead of “stated”

Sentence in 205-207: Please check grammar.

Figure 3 and 5: Again, same or similar experiments representing different time points (1 day and 6 weeks). Data should be combined to clearly show trends.

The authors use frequently “at six weeks after CSDS”, which can be changed often to six weeks/two weeks/1 day after CSDS.

Comments on the Quality of English Language

I outlined a few problems with the quality of English in my comments.

Author Response

In this manuscript the authors investigated “the effects of 90 ELF-EMF on behaviors, mitochondrial ETC proteins and activities in the brain, and oxidative stress responses in the brain in a mouse model of CSDS”. Their results show a beneficial effect on a number of mitochondrial functions.

Comment 1-1. According to the methods, the aggressor mouse was 10 weeks and older compared to 6-7 weeks of experimental mice. My question is whether the aggressor mouse is of the same strain or not? If not, why was a different mouse strain chosen, as this alone is already a significant stress factor for the mice?

Response 1-1. We appreciate valuable comments. The aggressor mouse was ICR strain, which was different from the experimental C57BL/6N mice. When we make a mouse model of depression by CSDS, ICR mice not C57BL/6 mice have been commonly used as an aggressor [1, 2]. We suppose this is because the ICR strain is more aggressive than the C57BL/6 strain.

Comment 1-2. Line 102: Is here meant, that both groups of mice have a similar SI ratio?  If yes, please change “became” to “were”.  Please consider the same in line 106.

Response 1-2. We appreciate a valuable suggestion. As suggested, we changed “became” to “were”.

Comment 1-3. Figure 1A shows which behavioral and biochemical experiment was done at a certain time point. Why where not all experiments done for all time points? It would be interesting to see the changes of behavior over time.

Response 1-3. Thank you for your thoughtful comments. We agree that temporal profiles of behavioral and biochemical features are of great interest. However, we hope that the reviewer understands that the analysis of 4 groups of 12 mice needed a lot of times and efforts. Biochemical assays were performed 6 weeks after CSDS, because they could be performed only after being sacrificed. We performed the three behavioral tests in some mice 2 and 6 weeks after CSDS. However, the effect of ELF-EMF was not prominent in the social interaction test 2 weeks after CSDS. The effect of CSDS disappeared in the tail suspension test and the forced swimming test 6 weeks after CSDS. We added relevant statements in Materials and Methods.

Comment 1-4. Figure 1C and D: this is an amazing result. I would have expected that already stressed and depressed mice want to get away from a stressful environment as quickly and as soon as possible. However, it appears that the mice gave up and adapted to a stressful situation more easily.

Response 1-4. Thank you for your comment. Both the tail suspension test and the forced swim test are designed to evaluate movements of mice when they are in an inescapable environment. These tests are widely used for assessing depressive states and the effects of antidepressants in rodents [3].

Comment 1-5. Please add in Figure 1, which results is from 1 day, 2 weeks or 6 weeks after CSDS.

Response 1-5. We appreciate the comment. As suggested, we indicated when the experiments were performed in Figure 1C, D, and F.

Comment 1-6. Figure 2 and 4: please combine the results of these figures. They show the same experiments for 2 different time points. Besides the expression levels of the ETC proteins,s it would be interesting to see their enzymatic activity. Expression may not change, but modification by acetylation, as the authors show in figure 4C, may change the activity.

Response 1-6. We appreciate valuable suggestions. As suggested, we combined Figures 2 and 4, and made new Figure 2. Measurement of the enzymatic activities of all ETC complexes requires a large amount of brain samples, which could be performed only by pooling multiple mouse brain samples. Instead, we measured oxygen consumption rate (OCR), which enabled the measurement of mitochondrial ETC activities in various conditions. The levels of spare respiratory capacity (Phase IV – Phase I) of OCR is reported to be maintained by Sirt3, which deacetylates mitochondrial ETC components, in cardiomyocytes [4]. We added statements in Materials and Methods to explain the measurement of individual ETC complexes were technically difficult.

Comment 1-7. Line 202: typo “sated” instead of “stated”

Response 1-7. Thank you for pointing this out. We corrected the typo.

Comment 1-8. Sentence in 205-207: Please check grammar.

Response 1-8. Thank you for pointing this out. We revised the indicated statement.

Comment 1-9. Figure 3 and 5: Again, same or similar experiments representing different time points (1 day and 6 weeks). Data should be combined to clearly show trends.

Response 1-9. We appreciate valuable suggestions. As suggested, we combined Figures 3 and 5, and made new Figure 3.

Comment 1-10. The authors use frequently “at six weeks after CSDS”, which can be changed often to six weeks/two weeks/1 day after CSDS.

Response 1-10. We appreciate the comment. We changed the expression.

References

  1. Golden, S. A.; Covington, H. E., 3rd; Berton, O.; Russo, S. J., A standardized protocol for repeated social defeat stress in mice. Nat. Protoc. 2011, 6, (8), 1183-91.
  2. Golden, S. A.; Covington, H. E., 3rd; Berton, O.; Russo, S. J., Corrigendum: a standardized protocol for repeated social defeat stress in mice. Nat. Protoc. 2015, 10, (4), 643.
  3. Belovicova, K.; Bogi, E.; Csatlosova, K.; Dubovicky, M., Animal tests for anxiety-like and depression-like behavior in rats. Interdiscip. Toxicol. 2017, 10, (1), 40-43.
  4. Pfleger, J.; He, M.; Abdellatif, M., Mitochondrial complex II is a source of the reserve respiratory capacity that is regulated by metabolic sensors and promotes cell survival. Cell Death Dis. 2015, 6, (7), e1835.

Reviewer 2 Report

Comments and Suggestions for Authors

The manuscript entitled: "Extremely low-frequency electromagnetic field (ELF-EMF) increases mitochondrial electron transport chain activities and ameliorates depressive behaviors in mice" by Teranishi et al. investigates the influence of low-frequency magnetic field on depressive-like behaviour in mice and links the behavioural amelioration with the properties of mitochondria within prefrontal cortex neurons. The study is very interesting and, in my opinion, has a high translational potential, however, I have a few questions and comments which I listed below:

1) there are several studies indicating that even as simple action as handling can affect the behavioural experiments in mice. In the study the authors performed FST two hours after TST. In our laboratory practice the behavioural tests are performed in at least 24 h intervals - in between the animals rest in their living cages to reduce the stress provoked by the previous procedure. In the study, the mice used for FST are just right after the TST which is quite stresfull condition, thus, the FST might not precisely refletcs the changes between the groups due to initial stress the animals are under. It is of great importnace, especially since the mice within the groups react differently on stresful conditions. I would like to ask - why the authors decided to performed FST two hours after the TST?

2) there is a general problem with the blots (especially in figures 2 and 4). The bands presented in the figures do not reflect the outcome from the graphs with quantification (e.g. in Fig. 3A the SOD2 level seems to be increased in CSDS mice). Despite the fact that WB is not 100% quantitative method (a subtle changes might be lost in WB), I would recommend to reconsider changing the representative images of WB. If there is a problem with obtaining a prominent differences in WB, maybe it would be worth to play with primary antibodies dillution. Additionally, some quantitative data regarding the blots were analyzed with Students t-test - thus, I have to ask, if the normality of data distribution was checked?

3) in CSDS mice the electrone transport chain seems to be affected: the proteins related to complexes III and IV are decreased. How the authors explain lack of OCR differences between CSDS mice and controls?

4) in both, control animals treated with EMF and CSDS mice after EMF treatment, the titer of SIRT3 and FOXO3A increases, however, only in CSDS-EMF mice this elevation is reflected by the increased SOD2 titer. How the authors explain the lack of SOD2 increase in control mice after ELF-EMF?

Author Response

The manuscript entitled: "Extremely low-frequency electromagnetic field (ELF-EMF) increases mitochondrial electron transport chain activities and ameliorates depressive behaviors in mice" by Teranishi et al. investigates the influence of low-frequency magnetic field on depressive-like behaviour in mice and links the behavioural amelioration with the properties of mitochondria within prefrontal cortex neurons. The study is very interesting and, in my opinion, has a high translational potential, however, I have a few questions and comments which I listed below:

Comment 2-1. There are several studies indicating that even as simple action as handling can affect the behavioural experiments in mice. In the study the authors performed FST two hours after TST. In our laboratory practice the behavioural tests are performed in at least 24 h intervals - in between the animals rest in their living cages to reduce the stress provoked by the previous procedure. In the study, the mice used for FST are just right after the TST which is quite stressful condition, thus, the FST might not precisely reflects the changes between the groups due to initial stress the animals are under. It is of great importance, especially since the mice within the groups react differently on stressful conditions. I would like to ask - why the authors decided to performed FST two hours after the TST?

Response 2-1. We appreciate valuable comments. We agree that serial FST and TST in a day would give an additional stress to our mice. However, serial FST and TST were also performed in previous reports [1-4]. We cited these articles in Materials and Methods.

Comment 2-2. There is a general problem with the blots (especially in figures 2 and 4). The bands presented in the figures do not reflect the outcome from the graphs with quantification (e.g. in Fig. 3A the SOD2 level seems to be increased in CSDS mice). Despite the fact that WB is not 100% quantitative method (a subtle changes might be lost in WB), I would recommend to reconsider changing the representative images of WB. If there is a problem with obtaining a prominent differences in WB, maybe it would be worth to play with primary antibodies dilution. Additionally, some quantitative data regarding the blots were analyzed with Students t-test - thus, I have to ask, if the normality of data distribution was checked?

Response 2-2. We appreciate valuable suggestions. As suggested, we substituted WB blots images in Figs. 2C, 2J, and 3C to represent quantitative analyses (we reorganized figures and the figure numbers became different). As suggested, we used the Shapiro-Wilk test followed by Bonferroni correction to examine the normal distributions of our datasets and found that SDHB in Fig. 2K had the lowest adjusted p-value of 0.183. We thus used parametric analyses throughout our manuscript. We added relevant statement in Materials and Methods.

Comment 2-3. In CSDS mice the electron transport chain seems to be affected: the proteins related to complexes III and IV are decreased. How the authors explain lack of OCR differences between CSDS mice and controls?

Response 2-3. Thank you for your thoughtful comments. We analyzed the expressions of mitochondrial ETC proteins in the prefrontal cortex, whereas the OCR was measured using the cerebral cortex. This was likely why we observed the difference. We added relevant statements in Results.

Comment 2-4. In both, control animals treated with EMF and CSDS mice after EMF treatment, the titer of SIRT3 and FOXO3A increases, however, only in CSDS-EMF mice this elevation is reflected by the increased SOD2 titer. How the authors explain the lack of SOD2 increase in control mice after ELF-EMF?

Response 2-4. Thank you for your scrutinizing comments. As suggested, in control mice, ELF-EMF increased Sirt3 and FoxO3A followed by a marginal increase of SOD2, and minimally decreased 4-HNE. However, the inhibition of complex II by ELF-EMF was not sufficient to markedly increase SOD2 and subsequently decrease 4-HNE. In contrast, CSDS alone induced mitochondrial oxidative stress (Fig. 3F). An additional inhibition of complex II by ELF-EMF was likely to have enhanced the oxidative stress and increased SOD2, which subsequently decreased 4-HNE. However, the actual mechanisms remain to be elucidated. We added a relevant statement in Results.

References

  1. Ma, M.; Ren, Q.; Yang, C.; Zhang, J. C.; Yao, W.; Dong, C.; Ohgi, Y.; Futamura, T.; Hashimoto, K., Adjunctive treatment of brexpiprazole with fluoxetine shows a rapid antidepressant effect in social defeat stress model: Role of BDNF-TrkB signaling. Sci. Rep. 2016, 6, 39209.
  2. Tang, J.; Xue, W.; Xia, B.; Ren, L.; Tao, W.; Chen, C.; Zhang, H.; Wu, R.; Wang, Q.; Wu, H.; Duan, J.; Chen, G., Involvement of normalized NMDA receptor and mTOR-related signaling in rapid antidepressant effects of Yueju and ketamine on chronically stressed mice. Sci. Rep. 2015, 5, 13573.
  3. Ouyang, X.; Wang, Z.; Luo, M.; Wang, M.; Liu, X.; Chen, J.; Feng, J.; Jia, J.; Wang, X., Ketamine ameliorates depressive-like behaviors in mice through increasing glucose uptake regulated by the ERK/GLUT3 signaling pathway. Sci. Rep. 2021, 11, (1), 18181.
  4. Chu, R.; Lu, Y.; Fan, X.; Lai, C.; Li, J.; Yang, R.; Xiang, Z.; Han, C.; Tian, M.; Yuan, H., Changes in SLITRK1 Level in the Amygdala Mediate Chronic Neuropathic Pain-Induced Anxio-Depressive Behaviors in Mice. J. Integr. Neurosci. 2024, 23, (4), 82.

Reviewer 3 Report

Comments and Suggestions for Authors

1)        Are there references or previous experiments by the authors about the production of reactive oxygen species in mitochondria in the CSDS model? This is to justify the presence of oxidative stress and because the only increase in 4-HNE does not explain increases of ROS in mitochondria.

2)        In the concept of hormesis during CSDS, what would be the molecular pathway involved in activating the Sirt3? 

3)        In lines 90-92, please add a paragraph including that mitophagy and dynamics were also evaluated in the study.

4)        Figure 1A is not cited in the main text.

5)        Please add a brief paragraph in the introduction about the relationship between ETS alterations and the development of fission and mitophagy processes in brain disease.

6)        The authors should clarify and provide references for why they use 1 day after CSDS and 6 weeks after CSDS. What would they expect in longer-term experiments?

7)        Figure 1E should say “measurement”

8)        In the introduction, please clarify how the extremely low-frequency electromagnetic field (ELF-EMF) increases mitochondrial ETC activity and hormesis?

Author Response

Comment 3-1. Are there references or previous experiments by the authors about the production of reactive oxygen species in mitochondria in the CSDS model? This is to justify the presence of oxidative stress and because the only increase in 4-HNE does not explain increases of ROS in mitochondria.

Response 3-1. We appreciate the comments. The elevation of mitochondrial ROS in the PFC of CSDS mice was previously reported (Fig. 1s and Fig. 4j in ref. [1]). We cited this article and added a relevant statement in Results.

Comment 3-2. In the concept of hormesis during CSDS, what would be the molecular pathway involved in activating the Sirt3?

Response 3-2. We appreciate the comments. We recently showed that ELF-EMF increased HSP70 in cultured cells [2]. Two possible mechanisms were speculated. First, as HSP70 induces Sirt3 [3], Sirt3 might have been induced via the activation of HSP70. Second, Sirt3 directly deacetylates SDHA and enhances its activity [4]. As ELF-EMF inhibits the activity of succinate dehydrogenase, Sirt3 might have been compensatorily induced to deacetylates and activates SDHA. We added a relevant paragraph in Discussion.

Comment 3-3. In lines 90-92, please add a paragraph including that mitophagy and dynamics were also evaluated in the study.

Response 3-3. We appreciate the comments. We added a suggested statement in Introduction.

Comment 3-4. Figure 1A is not cited in the main text.

Response 3-4. Thank you for pointing this out. We cited Figure 1A at the beginning of the first paragraph in Results.

Comment 3-5. Please add a brief paragraph in the introduction about the relationship between ETS alterations and the development of fission and mitophagy processes in brain disease.

Response 3-5. We appreciate the comments. As suggested, we added relevant statements in Introduction.

Comment 3-6. The authors should clarify and provide references for why they use 1 day after CSDS and 6 weeks after CSDS. What would they expect in longer-term experiments?

Response 3-6. Thank you for your comment. CSDS studies were terminated either at 2 [5], 4 [6], or 6 [7] weeks in previous studies. We performed the three behavioral tests in some mice 2 and 6 weeks after CSDS. However, the effect of ELF-EMF was more prominent in the social interaction test at 2 weeks than 6 weeks. The effect of CSDS disappeared in the tail suspension test and the forced swimming test at 6 weeks. We added relevant statements in Materials and Methods.

Comment 3-7. Figure 1E should say “measurement”

Response 3-7. Thank you for pointing out our inadvertent mistake. We corrected the typo.

Comment 3-8. In the introduction, please clarify how the extremely low-frequency electromagnetic field (ELF-EMF) increases mitochondrial ETC activity and hormesis?

Response 3-8. We appreciate the comment. As suggested, we added relevant statements in Introduction.

References

  1. Dong, W. T.; Long, L. H.; Deng, Q.; Liu, D.; Wang, J. L.; Wang, F.; Chen, J. G., Mitochondrial fission drives neuronal metabolic burden to promote stress susceptibility in male mice. Nat Metab 2023, 5, (12), 2220-2236.
  2. Huang, Z.; Ito, M.; Zhang, S.; Toda, T.; Takeda, J. I.; Ogi, T.; Ohno, K., Extremely low-frequency electromagnetic field induces acetylation of heat shock proteins and enhances protein folding. Ecotoxicol Environ Saf 2023, 264, 115482.
  3. Hu, B.; Wang, P.; Zhang, S.; Liu, W.; Lv, X.; Shi, D.; Zhao, L.; Liu, H.; Wang, B.; Chen, S.; Shao, Z., HSP70 attenuates compression-induced apoptosis of nucleus pulposus cells by suppressing mitochondrial fission via upregulating the expression of SIRT3. Exp. Mol. Med. 2022, 54, (3), 309-323.
  4. Finley, L. W.; Haas, W.; Desquiret-Dumas, V.; Wallace, D. C.; Procaccio, V.; Gygi, S. P.; Haigis, M. C., Succinate dehydrogenase is a direct target of sirtuin 3 deacetylase activity. PLoS One 2011, 6, (8), e23295.
  5. Dang, R.; Wang, M.; Li, X.; Wang, H.; Liu, L.; Wu, Q.; Zhao, J.; Ji, P.; Zhong, L.; Licinio, J.; Xie, P., Edaravone ameliorates depressive and anxiety-like behaviors via Sirt1/Nrf2/HO-1/Gpx4 pathway. J. Neuroinflammation 2022, 19, (1), 41.
  6. Warren, B. L.; Vialou, V. F.; Iniguez, S. D.; Alcantara, L. F.; Wright, K. N.; Feng, J.; Kennedy, P. J.; Laplant, Q.; Shen, L.; Nestler, E. J.; Bolanos-Guzman, C. A., Neurobiological sequelae of witnessing stressful events in adult mice. Biol. Psychiatry 2013, 73, (1), 7-14.
  7. Zhang, F.; Yuan, S.; Shao, F.; Wang, W., Adolescent Social Defeat Induced Alterations in Social Behavior and Cognitive Flexibility in Adult Mice: Effects of Developmental Stage and Social Condition. Front. Behav. Neurosci. 2016, 10, 149.

Round 2

Reviewer 2 Report

Comments and Suggestions for Authors

The authors have addressed all my questions and comments. I have no further concerns.

Just one more comment to the animal tests - the fact that some groups published a protocol regarding mice treating in tests and between tests does not mean that the protocol cannot be improved.

I recommend the manuscript to be published.

Author Response

Comment 2-1. The authors have addressed all my questions and comments. I have no further concerns.

Just one more comment to the animal tests - the fact that some groups published a protocol regarding mice treating in tests and between tests does not mean that the protocol cannot be improved.

I recommend the manuscript to be published.

Response 2-1. We appreciate your suggestion. As suggested, we revised our statements in Materials and Methods. We pointed out that previous works and our work with two serial tests in a day have drawbacks.

Revised statements in Section 4.6. The forced swim test was performed 2 h after the tail suspension test. Although the serial forced swim and tail suspension tests in a day were previously reported [112-115], the serial tests might have caused additional stress.

Reviewer 3 Report

Comments and Suggestions for Authors

The authors still misspelled the word “measurements” in Figure 1. My doubts were resolved and I have no additional comments.

Comments on the Quality of English Language

The authors still misspelled the word “measurements” in Figure 1. My doubts were resolved and I have no additional comments.

Author Response

Comment 3-1. The authors still misspelled the word “measurements” in Figure 1. My doubts were resolved and I have no additional comments.

Response 3-1. Thank you for pointing this out. We corrected the typo in Figure 1E.